# Interfering with Color Response by Porphyrin-Related Compounds in the MTT Tetrazolium-Based Colorimetric Assay

**DOI:** 10.3390/ijms24010562

**Published:** 2022-12-29

**Authors:** Bo Hee Choi, Mi-Ri Kim, Yu Na Jung, Smee Kang, Jungil Hong

**Affiliations:** Division of Applied Food System, College of Natural Science, Seoul Women’s University, 621 Hwarangro, Nowon-Gu, Seoul 01797, Republic of Korea

**Keywords:** porphyrin derivatives, MTT assay, interfering factors, tetrazolium, formazan decomposition, photosensitivity, photostability

## Abstract

Porphyrin compounds are widely distributed in various natural products and biological systems. In this study, effects of porphyrin-related compounds including zinc protoporphyrin (ZnPP), protoporphyrin IX (PPIX), cyanocobalamin (CBL), hemin, and zinc phthalocyanine (ZnPC) were analyzed on color response of 3-(4,5-dimethylthiazol-2-yl)-2,5-diphenyltetrazolium bromide (MTT) tetrazolium-based assay, a commonly-used method for analyzing cell viability. Color responses of MTT formazan formed in cells treated with ZnPP, PPIX, or ZnPC were significantly reduced even at submicromolar concentrations without affecting cell viability, whereas hemin and CBL did not. ZnPP, PPIX, and ZnPC rapidly induced degradation of MTT formazan already-produced by cells when exposed to light, but not under a dark condition. Photosensitizing properties of the three compounds were also verified through extensive generation of reactive oxygen species under light. The porphyrins did not affect the stability of water-soluble formazans including XTT, WST-1, WST-8, and MTS formazans. Several factors including different light sources and antioxidants modulated the degradation process of MTT formazan by the porphyrins. The results suggest that certain porphyrin compounds could cause a severe artifact in the MTT assay through rapid degradation of formazan dye due to their photosensitizing property, which needs to be considered carefully in the related assays.

## 1. Introduction

Analyzing methods using tetrazolium salts are commonly applied for investigating various biological events based on the determination of formazans, chromogenic dyes, formed through the reduction of tetrazolium by living cells. The reduction process of tetrazolium to the corresponding formazan occurs in living cells through the actions of reductases and dehydrogenases, which are related to many cellular activities. Among these assays, MTT [3-(4,5-dimethylthiazol-2-yl)-2,5-diphenyltetrazolium bromide] has been most widely-used to assess cell viability and cell proliferation [1,2,3]. The ring structure of MTT tetrazolium is cleaved through the reduction process mainly by mitochondrial dehydrogenase of living cells, and the tetrazolium is converted to MTT formazan with blue-purple color [4,5]. Water-insoluble MTT formazan is then dissolved using an organic solvent such as dimethyl sulfoxide (DMSO); viable cell numbers, rate of cell proliferation, or physiological state of cells can be evaluated quantitatively by the measurement of the color response of the formazan dye produced [5]. Several water-soluble formazan dyes have also been developed and applied for the experiments with a similar purpose [6]. These assays are relatively accurate and convenient for evaluating the cellular events related to cell growth and viability, and a large number of samples can be measured simultaneously using a multi-well plate, their application continues to expand in research using cells [3].

Porphyrins are a group of naturally occurring hetero- and macro-cyclic molecules consisting of 4 pyrrole subunits. They are widely distributed in living organisms and regulate important processes for maintaining life in bacteria, plants, animals, and human including light harvesting, oxygen storage and transport, electron transfer, and serving as cofactors for many enzymes [7]. Porphine, a basic form of porphyrins, has a highly-conjugated structure that exhibits strong light absorption property in the visible wavelength region of light [8]. Many physiological actions of porphyrins have been reported including antioxidant, anti-inflammatory, anti-obese, and anti-cancer activities [9]. Some animal studies have also indicated that treatment of porphyrin compounds showed beneficial effects for alleviating the health problems such as inflammation and diabetes [10]. Interestingly, the photodynamic therapy using porphyrins has been developed to induce cytotoxicity against cancer or diseased cells or to modulate their cellular activity, since they could convert absorbed light energy into chemical energy, affecting various cellular events in light-illuminated sites [11,12]. Phthalocyanine, a synthesized porphyrin-like derivative containing 4 isoindole units instead of pyrroles, has also been applied in photodynamic therapy [13]. This compound has the advantage of absorbing longer wavelength of light that can penetrate deeper into tissues and cells [14]. Porphyrins incorporating certain metal ions such as zinc protoporphyrin (ZnPP), a heme metabolite, have also been widely used as an inhibitor for heme oxygenase-1 (HO-1) in the related studies [15]. Accordingly, application of porphyrins in studies using cells and biological systems has become prevalent [16].

A tetrazolium assay is performed by measuring color intensity of formazan produced through the reduction process by cells. Therefore, various factors can affect the color response of the assay, including condition of cells, chemical features of compounds treated, and composition of media that interfere with the reduction process of tetrazolium [5,17,18]. Previously, we observed the dramatic decrease of color response of MTT formazan in ZnPP-treated cells without affecting viability of the cell [19]. In the present study, we tried to extend our knowledge through investigating the actions of porphyrin-containing derivatives including protoporphyrin IX (PPIX), hemin, cyanocobalamin (vitamin B12, CBL) as well as zinc phthalocyanine (ZnPC), a porphyrin-related compound in the MTT assay (Appendix A). Their mechanisms involved in modulating color response in the MTT assay, and effects of the porphyrin compounds on different tetrazolium-based assays were also investigated.

## 2. Results and Discussion

### 2.1. Effects of Porphyrin Derivatives on Cell Viability Measured by the MTT Assay

Biological effects of various natural compounds and drugs including porphyrin derivatives on growth and viability of cells have been assessed using several tetrazolium-based assays [20]. Among them, the MTT tetrazolium-based method has been most-commonly applied. Previously, we reported that significant color reduction of MTT formazan in cells treated with ZnPP, which was a serious interfering factor for this experimental assay [19]. In the present study, modulation of color response of formazan produced in the assays using MTT or other different tetrazolium salts by several porphyrin-related compounds was investigated. 

INT 407 cells were treated with five porphyrin-related derivatives including ZnPP, PPIX, ZnPC, hemin, and CBL for 24 h, and the cell viability was analyzed using the MTT assay. Among the compounds, color response of MTT formazan of cells treated with ZnPP, PPIX, and ZnPC decreased noticeably even at 1 μM, whereas apparent viability cells treated with hemin or CBL was not changed significantly up to 10 μM (Figure 1A). Sub-micromolar levels of ZnPP, PPIX, and ZnPC also significantly reduced the color response in the MTT assay, but there was no clear dose-dependent effect shown especially in the cells treated with ZnPC (Figure 1B). Apparent viabilities of the cells incubated with ZnPP, PPIX, and ZnPC for 48 h were rather increased compared to those of the cells treated for 24 h; this property was different that of typical cytotoxic compounds (Figure 1C). Interestingly, the number of living cells analyzed using the trypan blue assay was not changed significantly compared to the control (Figure 1D). Morphology of these cells treated with the 3 compounds (10 μM) for 24 h did not show a drastic change even in cells showing less than 50% survival based on the MTT assay either (Appendix A). 

An organic solvent such as DMSO is commonly used to solubilize MTT formazan formed in cells. After adding DMSO to cells, a time period of several minutes for homogeneously dissolving formazan formed in the cells is often applied, and the absorbance of formazan dissolved is then measured; the dissolution time was set for ~5 min in the current experiment. In the subsequent experiment, the absorbance measurement was performed immediately by minimizing this time to within 1 min. As a result, there was marginal difference observed in the apparent viability of cells treated with ZnPP. The cells treated with PPIX or ZnPC still exhibited a significant decrease in cell viability, although to a lesser extent (Figure 2A). From these observations, we suspected that the time interval from DMSO addition for formazan dissolution to absorbance measurement was an important factor in the decrease in formazan color intensity. We, next, evaluated the formazan levels at different measurement time intervals after dissolving formazan formed in cells treated with ZnPP, PPIX or ZnPC for 24 h. When measured immediately after formazan was dissolved in DMSO, all cells treated with different concentrations of ZnPP (0.5–10 μM) showed a similar formazan color intensity with ~95% compared to control (Figure 2B). In the case of PPIX-treated cells, the degree of formazan color intensity was 72.6–86.1% depending on PPIX treatment concentration, whereas ZnPC-treated cells showed less than 70% of formazan color development, regardless of the difference in treatment concentration (Figure 2C,D). The color reduction of MTT formazan was more prominent as the time interval and the treatment concentration increased. The decrease in absorbance was most rapid in the cells treated with ZnPC, and the amount of residual formazan after 60 min in cells treated with 5–10 μM dropped to less than 10%. (Figure 2D). Formazan in cells treated with 5–10 μM of hemin or CBL, however, did not show a significant decrease in color response even when measured after 60 min of DMSO addition (Figure 2E). We also used 0.2 N NaOH as a solvent for solubilizing formazan formed in cells after incubation with ZnPP for 24 h, and the results indicated a similar pattern of formazan degradation to that of DMSO at a much faster rate (Figure 2F).

### 2.2. Effects of Porphyrin-Related Compounds in Different Tetrazolium Based-Assays

Since the 3 porphyrin derivatives reduced apparent cell viability by decreasing color response of MTT formazan, different assay methods for evaluating their effects on cell viability were applied. The MTT assay has a disadvantage of extracting water-insoluble MTT formazan formed inside cells using an organic solvent such as DMSO [5]. To compensate for this shortcoming, several tetrazolium salts which can be converted to water-soluble formazans have been developed [6]. These water-soluble formazans are produced by reduction from tetrazolium present in extracellular medium, and electron coupling agents such as methoxy-5-methylphenazinium methyl sulfate (PMS) or menadione are used to link intracellular reductase activity with the reduction of tetrazolium in medium [20]. For measuring such water-soluble formazans, it is not necessary to use an organic solvent for formazan extraction, and cell damages can also be avoided. 

INT 407 cells were treated with each porphyrin (10 μM) including ZnPP, PPIX, or ZnPC for 24 h, and XTT or MTS tetrazolium were applied for measuring cell viability (Figure 3A,B). According to the results from color intensity of the water-soluble formazans formed in media, no significant change by all the 3 porphyrin derivatives was observed. When measured after 60 min, XTT formazan produced by cells treated with PPIX and ZnPC decreased slightly by ~10%, whereas color intensity of MTS formazan was not significantly changed (Figure 3A,B). Other tetrazolium salts including WST-1 and WST-8 to be reduced to water-soluble formazans were also applied; there was no noticeable decreases in viability of cells treated with ZnPP, PPIX, or ZnPC at 10 μM. (Figure 3C,D). These results indicate that certain porphyrins and related compounds could decrease color response of MTT formazan without affecting cell viability. However, this phenomenon was not observed in the assays using the tetrazolium salts such as XTT, MTS, and WST, which is converted to water soluble formazans. 

### 2.3. Effects on Formation of MTT Tetrazolium in Cells

The main principle of tetrazolium-based assays is the formation of formazan through the reduction of tetrazolium in living cells [3,4,5]. However, the reductase activity in cells might change depending on the physiological state or culture environment of cells, which could affect the formazan production. Accordingly, we suspected that the 3 porphyrin-related compounds might affect formation of MTT formazan either through delay of MTT tetrazolium reduction or through reversible oxidation of the formazan formed in cells. 

In the consequent experiment, we investigated whether the porphyrin derivatives could affect a reduction rate of MTT tetrazolium to formazan in living cells. For the experiment, the same population of cells were treated with MTT tetrazolium and porphyrins (10 μM) simultaneously, and the amount of formazan formed was measured immediately after removing the treatment medium after different incubation periods (Figure 4A). The amount of formazan formed in cells increased linearly in proportion to the incubation time to the cells during 3 h. The simultaneous addition of the porphyrin compounds did not affect the reduction rate of MTT tetrazolium or the formation rate of MTT formazan in cells markedly up to 10 μM; only PPIX delayed the rate slightly (Figure 4B). This result indicates that the 3 compounds did not affect intracellular formazan formation and tetrazolium reduction but affect the stability of already formed and dissolved formazan in a solvent.

### 2.4. Effects of Porphyrins on Stability of MTT Formazan

The subsequent experiment evaluated the effects of the porphyrin-related compounds on the color stability of formazan produced by the cells and dissolved in DMSO. MTT formazan was allowed to produce by INT 407 cells without porphyrin treatment, and the cells containing formazan were lysed using DMSO. When each porphyrin (5 μM) was added to the cell lysates containing formazan, the newly added porphyrins induced a rapid decolorization of formazan, and the residual color intensity decreased to less than 10% within 1 h (Figure 5A). In the formazan lysates added with different concentrations of each porphyrin (0–5 μM), the decolorization of MTT formazan occurred in a concentration- and time-dependent manner (Figure 5B–D). Among the 3 porphyrin derivatives, ZnPP and ZnPC induced more rapid degradation; the decolorization rate was somewhat delayed by PPIX as compared to the Zn-containing porphyrin derivatives. In another experiment, we also tested the effects of Zn on formazan stability; the acceleration of formazan degradation by Zn was not observed (Appendix A). The similar pattern of decolorization was also observed in the experiment using a pure chemical formazan (Figure 5E). Interestingly, the degradation of MTT formazan by porphyrins was not induced under a light-blocked dark condition (Figure 5F). Effects of the three porphyrin-related compounds on the color intensity of water-soluble formazans produced from XTT, MTS, WST-1, and WST-8 tetrazolium were also analyzed. Their color intensities decreased slightly under light exposure; the porphyrin derivatives, however, did not accelerate degradation of the water-soluble formazans during 180 min of light exposure (Appendix A).

MTT formazan is known to be sensitive to light [21,22]. We also observed degradation of ~15% formazan produced by cells within 90 min under light exposure (Figure 5A). The addition of porphyrins, however, dramatically accelerated the degradation rate of MTT formazan under a typical laboratory light condition. These results suggest that certain porphyrin-related compounds cause profound effects on MTT formazan stability, and the degradation of MTT formazan by porphyrins is significantly accelerated upon light exposure. Therefore, light exposure must be strictly controlled when performing the MTT assay with porphyrins and related compounds.

### 2.5. Evaluation of Photo-Reactive Property of Porphyrin Derivatives

Our present study indicates that certain porphyrin derivatives strongly induced decolorization of MTT formazan, whereas hemin and CBL with a similar porphyrin ring structure did not affect the formazan stability. Many porphyrin-related compounds exhibit a light-sensitive property and cause chemical changes in neighboring molecules by either donating or extracting electrons under light [16,23]. Accordingly, these photosensitizing compounds frequently generate reactive oxygen species (ROS) from oxygen under irradiated conditions [24,25]. Accordingly, we hypothesized that the photosensitizing property of porphyrin compounds induce degradation of MTT formazan only under light, and the ones lacking this property do not have a potential for the light-induced degradation. 

Dichlorofluorescein diacetate (DCFH-DA) is widely used to measure intracellular ROS [26]. The DA moiety of DCFH-DA is cleaved and converted to DCFH by intracellular esterase, and DCFH reacts with ROS to convert to DCF, which emits strong green fluorescence [27]. After incubation of INT 407 cells with DCFH-DA, DCFH containing cell lysates were prepared. The cell lysates treated with each porphyrin derivative or methylene blue (MB), a known photosensitizer, for 24 h were mixed with the lysates containing DCFH. The mixtures were then stored in the dark for 60 min followed by exposure to light (Figure 6A). DCF fluorescence of all the mixtures was not changes during 60 min in the dark; fluorescence intensities of the mixtures containing ZnPP, PPIX, ZnPC as well as MB was markedly enhanced with irradiation time after light exposure. The mixture containing CBL and hemin, however, did not present a significant difference of DCF fluorescence from the control during 300 min. 

Effects of fresh porphyrins not treated with cells were also analyzed. Although there was somewhat difference in the order of activity from the cases of samples treated with cells, ZnPP, PPIX, and ZnPC still enhanced the DCF fluorescence significantly only after light exposure (Figure 6B); their fluorescence enhancing activity under light showed a concentration-dependent manner (Figure 6C). Fresh CBL and hemin did not show a photosensitizing property either. These results indicate that ZnPP, ZnPC, and PPIX increased the fluorescence intensity of DCF as they generated ROS under light conditions; these photo-reactive property might be importantly involved in decomposing MTT formazan. On the other hand, even porphyrin-related compounds without showing these properties such as CBL and hemin do not appear to have any effect on the formazan degradation. Interestingly, fresh ZnPP showed the most potent activity in enhancing DCF fluorescence, whereas PPIX had the strongest activity among the porphyrins treated with cells for 24 h. It might be due to the different metabolic features of porphyrins in cells [28]. 

### 2.6. Effects of Porphyrins on MTT Formazan Degradation under Different Light Sources

Photosensitizing property of certain porphyrin derivatives has been widely studied [3,12,13]. Among the porphyrin-related compounds used in the present study, ZnPP, PPIX, and ZnPC show photosensitizing properties, and these activities might play an important role for reducing MTT formazan stability under light. Photosensitizing property is expressed by the energy absorbed in a specific wavelength range of light, which is different for each compound [23,24]. It was observed that the porphyrin derivatives that induced MTT formazan decomposition in this study had somewhat different light absorption wavelength ranges. ZnPP and PPIX showed a maximum light absorption at the 410 and 400 nm region, respectively, whereas ZnPC had light absorption characteristics in the two regions of 350 and 670 nm (Figure 7A). In the following experiments, the formazan decolorization patterns by each porphyrin were investigated using various light sources such as a regular fluorescence light (visible region), UVC (254 nm), and UVA (365 nm). Under a general fluorescence light, the formazan decolorization by ZnPC occurred most rapidly, followed by ZnPP and PPIX (Figure 7B). When irradiated with UVC, ZnPP showed the strongest decolorization activity, and the overall formazan decolorization rate by the three porphyrin compounds was delayed compared to the cases under a fluorescence light (Figure 7C). UVA irradiation induced the most rapid decolorization of formazan by the three porphyrin derivatives showing more than 60% of formazan decomposition within 30 min (Figure 7D). However, CBL and hemin did not present notable effects on formazan decolorization under all three light sources.

The half-lives for decolorization of MTT formazan by the three porphyrin-related compounds indicate that 6–16 min under UVA were over 10 times faster than ones (88–114 min) under UVC (Figure 7E). Accordingly, the decomposition of MTT formazan by each compound proceeded more rapidly under UVA than under UVC with shorter wavelength and stronger energy. In particular, ZnPC with a light absorption region in a visible region (670 nm) showed a much faster formazan decomposition effect under a fluorescence light than UVC. These observations indicate that the formazan decolorization occurs effectively at the light absorption wavelength of each porphyrin. Therefore, the degradation of MTT formazan by the porphyrin-related compounds is attributable to their photosensitizing activity based on the absorbed energy from specific wavelength of light rather than the total amount of light energy exposed.

### 2.7. Effects of Different Antioxidants and Electron Coupling Agents 

Our results indicate that the porphyrin derivatives induced degradation of MTT formazan due to their photosensitizing property. It was expected that ROS formation under light appeared to be involved in this action. Accordingly, effects of different types of antioxidants including butylated hydroxytoluene (BHT), β-carotene, and *n*-acetylcysteine (NAC) on MTT formazan degradation by the porphyrins were investigated. 

As the most common antioxidant mechanism, free radical scavenging activity by phenolic compounds such as BHT has been reported [29,30,31]. However, BHT had little effect on delaying decolorization of MTT formazan by ZnPP. On the other hand, treatment with NAC [32] or β-carotene [33,34], which have singlet oxygen quenching property, significantly delayed MTT formazan degradation; the effect of NAC was more prominent (Figure 8A). The antioxidant compounds also induced in decrease of DCF fluorescence similar to the tendency to delay the decolorization of MTT formazan by ZnPP (Figure 8B). Since production of singlet oxygen plays an important role in the oxidation process by photosensitizers [35], it is believed that singlet oxygen produced from porphyrins under light is importantly involved in decolorization of MTT formazan.

We observed that the porphyrin derivatives little affected stability of water-soluble formazan (Figure 3, Appendix A). Since electron coupling agents such as menadione were frequently used in the related assays [20], effect of menadione on degradation of MTT formazan by ZnPP was also evaluated. Menadione significantly delayed the degradation of MTT formazan (Figure 8C); the result suggests that an electron coupling reagent in the assays using water-soluble formazans might be one of the reasons for stabilizing the formazans in the presence of porphyrins.

The MTT tetrazolium-based assay has been widely applied for evaluating cytotoxic effects of various agents, and many interfering factors in this assay were observed [5,17,18]. In the present study, we report that certain porphyrin-related compounds could induce a rapid degradation of MTT formazan under light exposure due to their photosensitizing property, resulting in significant artifacts in this assay. Accordingly, use of the MTT assay should be avoided when analyzing effects of porphyrin-related compounds and perhaps other photo-reactive compounds on cell viability. In particular, the present results should be carefully considered in the studies evaluating the cytotoxic efficacy of photodynamic agents that usually show photosensitizing property using a tetrazolium-based analysis.

## 3. Methods and Materials

### 3.1. Chemicals and Cell Lines

MTT tetrazolium was purchased from Amresco Inc. (Solon, OH, USA). Menadione and 2,3-bis(2-methoxy-4-nitro-5-sulfophenyl)-2H-tetrazolium-5-carbox-anilide (XTT) were from MPbio (Santa Ana, CA, USA). Cell viability assay kits using 2-(4-iodophenyl)-3-(4-nitrophenyl)-5-(2,4-disulfophenyl)-2H-tetrazolium (WST-1), 2-(2-methoxy-4-nitrophenyl)-3-(4-nitrophenyl)-5-(2,4-disulfophenyl)-2H-tetrazolium (WST-8), and 3-(4,5-dimethylthiazol-2-yl)-5-(3-carboxymethoxyphenyl)-2-(4-sulfophenyl)-2H-tetrazolium (MTS) were obtained from Daeil Lab Service (Seoul, Republic of Korea), Dojindo (Kumamoto, Japan), and Promega (Promega, Madison, WI, USA), respectively. Trypan blue was from GIBCO (Grand island, NY, USA). ZnPP, PPIX, ZnPC, CBL, hemin, and all other chemicals were from Sigma-Aldrich chemical Co. (St. Louis, MO, USA). INT 407 (HeLa derived) immortalized human intestinal cell line (CCL-6) was provided from America type culture collection (ATCC, Manassas, VA, USA). INT 407 cells were grown in MEM medium, supplemented with 10% fetal bovine serum, 100 unit/mL penicillin, 0.1 mg/mL streptomycin, and 1% non-essential amino acids. The cells were kept at 37 °C in 95% humidity and 5% CO_2_.

### 3.2. Analysis of Cell Viability

INT 407 cells were seeded in a 96-well plate at ~1.5 × 10^4^ cells. When the cells reached approximately 80% confluency, the cells were treated with different concentrations (0–10 μM) of each porphyrin compound for 24 h. After removing the medium containing the compounds treated, fresh serum free medium (100 μL) containing MTT tetrazolium (0.5 mg/mL) was added to each well. The cells were incubated for further 1 h at 37 °C. MTT formazan formed in cells was allowed to be dissolved with DMSO, and the absorbance was measured at 550 nm using a microplate reader at time intervals from 0 to 60 min (Triad LT; Dynex Technologies Inc., Chantilly, VA, USA). A medium containing XTT tetrazolium (0.1 mg/mL) and menadione (40 μM) were added to cells for 1 h and color intensity developed in the medium was analyzed directly. Analysis of cell viability using the WST-1, WST-8, and MTS tetrazolium kit was performed according to the manufacturer’s procedures. In addition, the number of living cells were counted using trypan blue staining after incubation of cells with each porphyrin-related compound for 24 h.

### 3.3. Analysis of Effects on the MTT Formazan Formation

To investigate effects of porphyrin-related compounds the reduction of MTT tetrazolium and formation of MTT formazan, INT 407 cells were treated with each compound (10 μM) and 0.5 mg/mL of MTT tetrazolium simultaneously and incubated at 37 °C for 180 min. At different time points, the culture medium containing a porphyrin compound was removed, and MTT formazan formed in cells was dissolved in DMSO. The absorbance was analyzed at 550 nm (Triad LT). All procedures of the experiment were performed in a dark condition.

### 3.4. Analysis for Changes in Color Stability of Different Formazans by Porphyrin Compounds

MTT formazan products were prepared from the incubation of MTT tetrazolium salt with INT 407 cells. The cells were seeded in a 10 cm dish until reached ~80% confluency, the cells were then treated with MTT tetrazolium for 2 h, and MTT formazan formed in cells was lysed using DMSO. Pure MTT formazan (Sigma-Aldrich Co.) dissolved in DMSO was also used. For preparation of water-soluble formazans, INT 407 cells grown in a 10 cm dish were incubated with XTT tetrazolium (0.1 mg/mL) and menadione (40 μM), or incubated with MTS tetrazolium reagents according to the manufacturer’s procedures in media for 3 h at 37 °C. The water-soluble formazans formed in media was collected after brief centrifugation (10,000× *g*) for 10 min. Each porphyrin dissolved in DMSO or PBS (10 μM, 50 µL) was added to an equal volume of the cell lysates containing MTT formazan or the media containing water-soluble formazans, respectively. The mixtures were incubated under a regular fluorescent light (TL-D Super 80 fluorescent lamp 32W, Philips, Seoul, Republic of Korea) or dark conditions at room temperature; their peak absorbance change was analyzed (Triad LT).

### 3.5. Evaluation of Photosensitizing Properties of Porphyrins

Dichlorofluorescein diacetate (DCFH-DA, 10 μM) was incubated with INT 407 cells for 1 h at 37 °C to produce DCFH in cells. The media were, then removed, and the cell lysates containing DCFH were prepared by adding DMSO. The cell lysate containing DCFH (50 µL) was mixed with an equal volume of porphyrin-related compounds dissolved in DMSO or INT 407 cell lysate treated with each porphyrin compound (10 μM) for 24 h. DCF fluorescence intensity of the reaction mixture under a fluorescent light (TL-D Super 80 fluorescent lamp) or a dark condition was analyzed at an emission 535 nm (excitation 485 nm) using a multiplate reader (SpectraMax M3, Molecular device, Sunnyvale, CA, USA).

### 3.6. Assessment of Various Factors Affecting MTT Formazan Decolorization by Porphyrins

The cell lysates containing MTT formazan (100 μL) reduced by INT 407 cells from MTT tetrazolium was incubated with each porphyrin compound (10 μM, 100 μL) in a 96 well plate under UVA (365 nm) and UVC (254 nm) using an UV irradiation equipment (230 V, 50/60 Hz) (VL-4LC; Vilber Lourmat, Marne-la-Vallée, France) with 8 cm distance or a fluorescent light (TL-D Super 80 fluorescent lamp) with 30 cm distance at room temperature. To investigate effects of several antioxidants or menadione, an electron coupling agent, on degradation of MTT formazan by ZnPP, the cell lysates containing MTT formazan (100 μL) and ZnPP (20 μM, 50 μL) with or without each antioxidant (50 μL) or menadione (160 μM, 50 μL) were incubated under a fluorescent light. The absorbance changes of formazan were measured at different time points at 550 nm (Triad LT).

### 3.7. Data Analysis

All values represent the mean ± standard deviation (SD). Statistical significance was evaluated by the two-tailed Student’s *t*-test. One-way ANOVA and the Tukey’s honestly significant difference (HSD) test were used for comparing multiple results (SPSS, Version 21.0, Chicago, IL, USA).

## Figures and Tables

**Figure 1 ijms-24-00562-f001:**
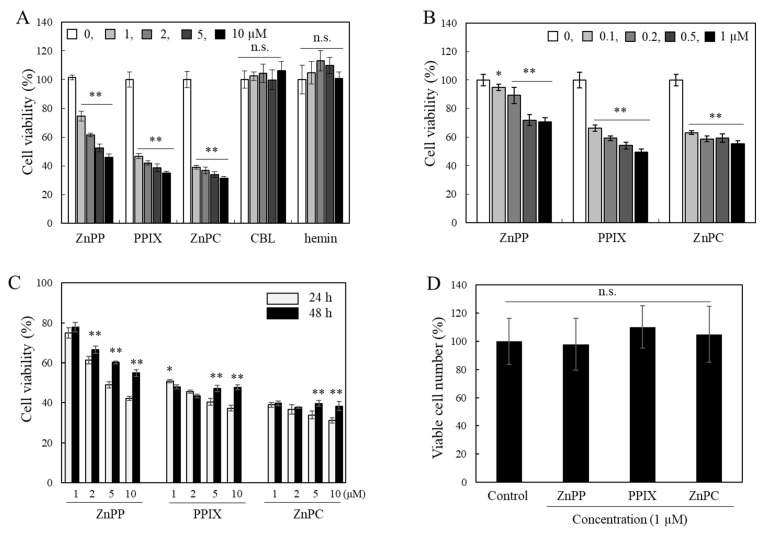
Effects of different porphyrin-related compounds on cell viability analyzed using the MTT assay. INT407 cells were treated with different concentrations of porphyrin-related compounds (0–10 µM) (**A**) or sub-micromolar levels (0–1 µM) (**B**) for 24 h, and cell viability was analyzed using the MTT assay. Effects of 3 porphyrin derivatives after 24 and 48 h treatment were compared (**C**). Numbers of viable cells using the trypan blue assay were also analyzed (**D**). Each value represents the mean ± S.D. (*n* = 8). *, ** Significantly different from control or between 24 and 48 h (in (**C**)) according to Student’s *t*-test (*, *p* < 0.05; **, *p* < 0.01). n.s., not significant.

**Figure 2 ijms-24-00562-f002:**
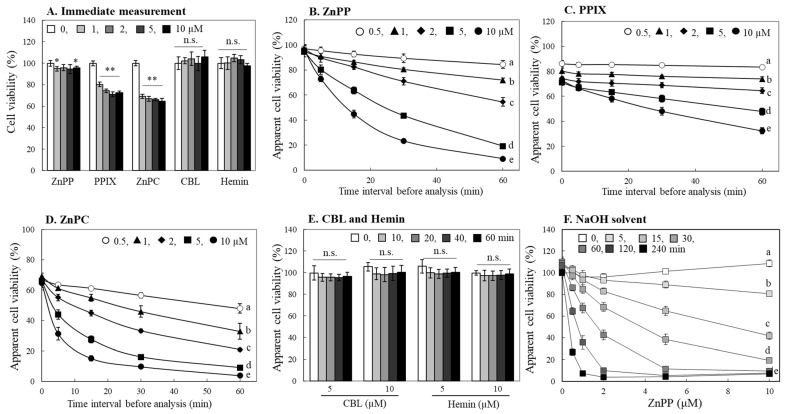
Changes in apparent viability of cells treated with porphyrin derivatives by the measurement time interval. INT407 cells were treated with different concentrations (0–10 µM) of porphyrin derivatives for 24 h. The MTT assay was, then, performed immediately (**A**) or by changing the time interval from the lysis of cells treated with ZnPP (**B**), PPIX (**C**) ZnPC (**D**) using DMSO to the absorbance measurement (0–60 min). The effects of CBL and hemin (5 and 10 µM) on apparent cell viability were also analyzed during 60 min of the time interval (**E**). Cells treated with different concentrations of ZnPP were lysed using 0.2 N NaOH and the formazan absorbance Dwas analyzed at different time interval (**F**). Each value represents the mean ± S.D. (*n* = 8). *, ** Significantly different from control according to Student’s *t*-test (*, *p* < 0.05; **, *p* < 0.01). Different letters indicate a significant difference among samples (*p* < 0.05) based on one way ANOVA and the Tukey’s HSD test. n.s., not significant.

**Figure 3 ijms-24-00562-f003:**
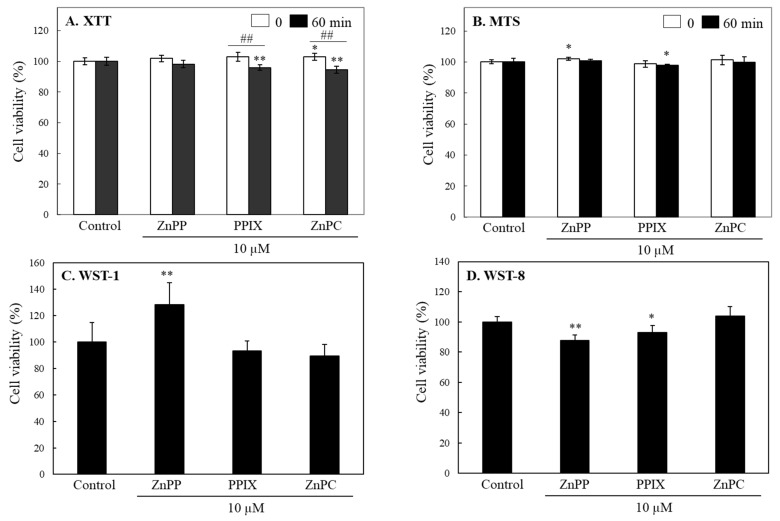
Effects of porphyrin derivatives on cell viabilities analyzed using different tetrazolium assays. INT407 cells were treated with each porphyrin derivatives (10 µM) for 24 h, and formazan formed by cells from XTT (**A**) and MTS (**B**) tetrazolium was analyzed at 0 and 60 min. Effects of three porphyrin derivatives on cell viability were also analyzed after 24 h treatment using WST-1 (**C**) and WST-8 (**D**) tetrazolium. Each value represents the mean ± S.D. (*n* = 6–8). *, ** Significantly different from control or ## between 0 and 60 min according to Student’s *t*-test (*, *p* < 0.05; **, ##, *p* < 0.01).

**Figure 4 ijms-24-00562-f004:**
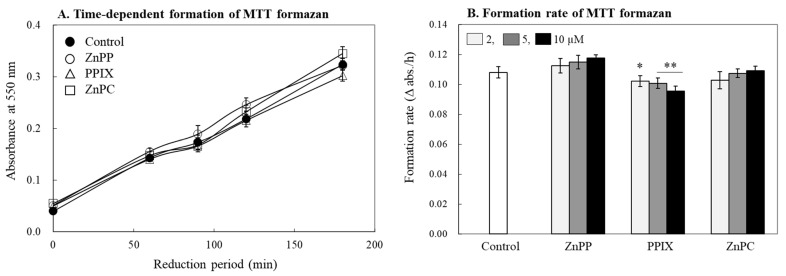
Effects of porphyrin derivatives on the formation of MTT formazan in cells. A same population of INT 407 cells were treated with ZnPP, PPIX, and ZnPC (10 µM) with MTT reagent simultaneously, and reduction of MTT tetrazolium to formazan in cells was measured at 550 nm during 3 h (**A**). The reduction rates (Δ abs./h) in the absence or presence of each porphyrin derivatives were also calculated (**B**). Each value represents the mean ± S.D. (*n* = 4). *, ** Significantly different from control according to Student’s *t*-test (*, *p* < 0.05; **, *p* < 0.01).

**Figure 5 ijms-24-00562-f005:**
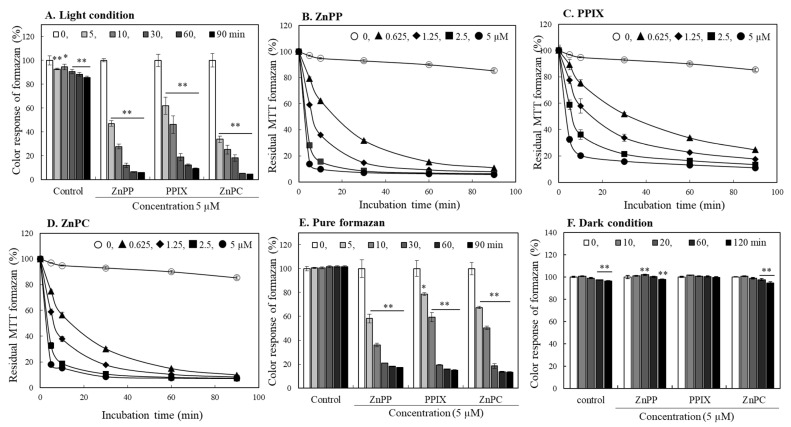
Effects of porphyrin derivatives on color stability of MTT formazan under light or dark conditions. MTT formazan produced by cells was dissolved in DMSO and was mixed with each porphyrin (5 µM). The mixture was irradiated at room temperature under a fluorescence light, and changes of color intensity of MTT formazan were analyzed at 550 nm during 90 min (**A**). Effects of ZnPP (**B**), PPIX (**C**), and ZnPC (**D**) on the color stability of MTT formazan produced by cells or pure formazan (**E**) were analyzed at different time points under light. Effects of each porphyrin (5 µM) on color stability of MTT formazan were also measured in a dark condition (**F**). Each value represents the mean ± S.D. (*n* = 8). *, **, Significantly different from control according to Student’s *t*-test (*, *p* < 0.01; **, *p* < 0.001).

**Figure 6 ijms-24-00562-f006:**
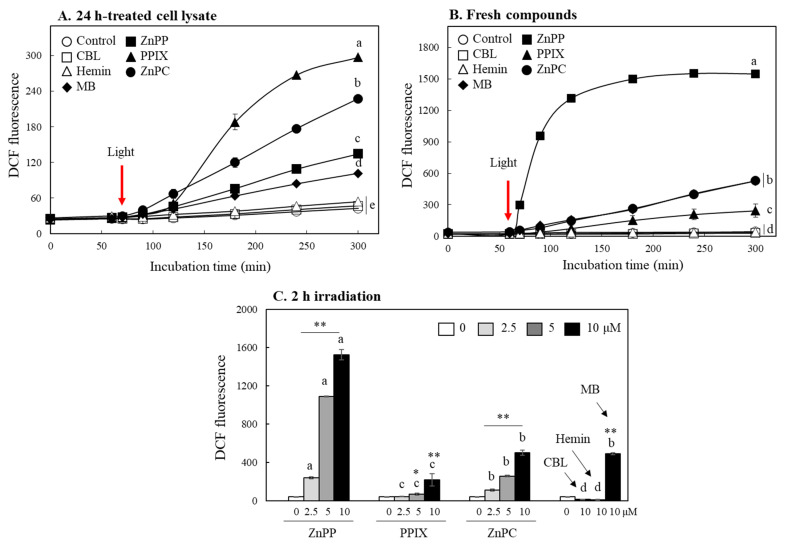
Evaluation of ROS generation property of different porphyrin derivatives under light using DCFH. DCFH prepared from DCFH-DA as described in materials and methods was mixed with INT 407 cell lysates treated with each porphyrin or MB (10 μM) for 24 h (**A**) or fresh compounds (each 5 μM) (**B**). The mixture was incubated in a dark for 60 min followed by a light condition, and changes in DCF fluorescence were analyzed during 6 h. Effects of different concentrations of porphyrins on DCF fluorescence changes during 2 h under light were also monitored (**C**). Each value represents the mean ± SD (*n* = 3). *, ** Significantly different from control according to Student’s *t*-test (*, *p* < 0.05; **, *p* < 0.01). Different letters indicate a significant difference among different porphyrins (**A**,**B**) or samples within same concentration (**C**) (*p* < 0.05) based on one way ANOVA and the Tukey’s HSD test.

**Figure 7 ijms-24-00562-f007:**
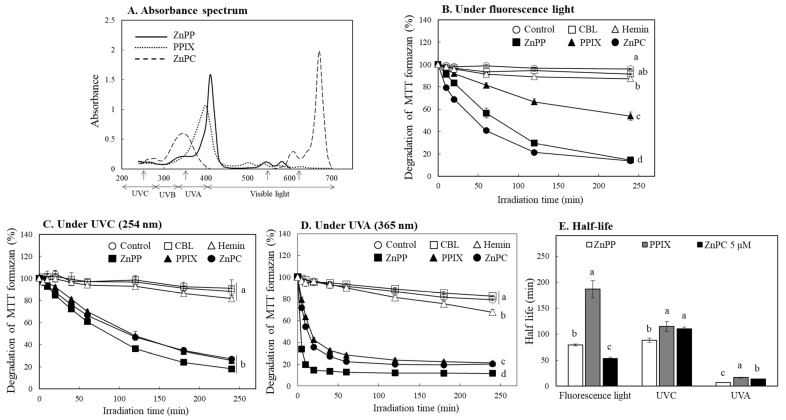
Effects of porphyrins on degradation of MTT formazan under different light conditions. Absorbance spectrum of ZnPP, PPIX, and ZnPC dissolved in DMSO (10 μM) was analyzed (**A**) (The upward arrows in A indicate the emission peaks of the light source used). MTT formazan formed was irradiated with different porphyrins under a fluorescent light (**B**), UVC (254 nm) (**C**) or UVA (365 nm) (**D**) during 4 h. Time-dependent changes in color response of MTT formazan in the presence of each compound (5 µM) were analyzed using a microplate reader at 550 nm. Half-lives of formazan color degradation by each porphyrin under different lights were also calculated (**E**). Each value represents the mean ± S.D. (*n* = 3). Different letters indicate a significant difference among different light treatments (*p* < 0.05) based on one way ANOVA and the Tukey’s HSD test.

**Figure 8 ijms-24-00562-f008:**
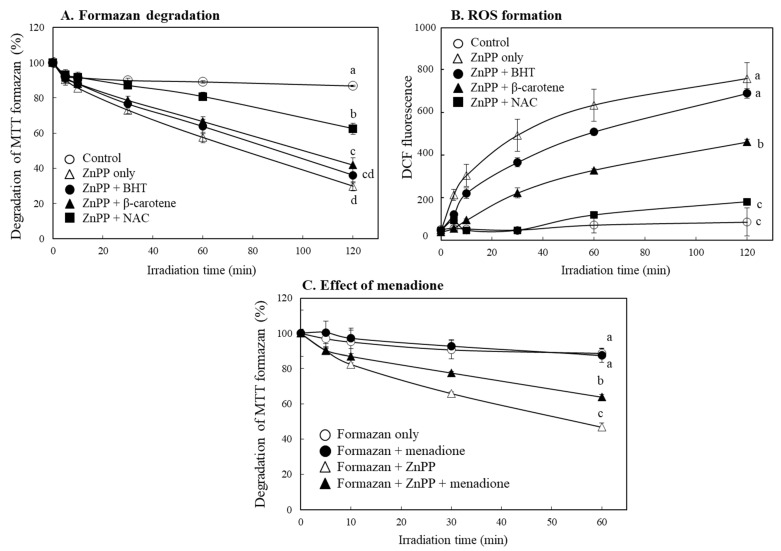
Effects of different antioxidants and menadione on MTT formazan degradation by porphyrins and DCF fluorescence. Effects of different antioxidants (each 25 µM) on the degradation of MTT formazan (**A**) and enhancement of DCF fluorescence (**B**) by ZnPP (5 µM) under light exposure were analyzed for 2 h. MTT formazan dissolved in DMSO was mixed with ZnPP (5 µM) and menadione (40 µM), an electron coupling agent, and color response of MTT formazan was also analyzed under light (**C**). Each value represents the mean ± SD (*n* = 3). Different letters indicate a significant difference among samples (*p* < 0.05) based on one way ANOVA and the Tukey’s HSD test.

## Data Availability

Data available from the corresponding author upon reasonable request.

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
