# Peer review of "Interfering with Color Response by Porphyrin-Related Compounds in the MTT Tetrazolium-Based Colorimetric Assay"

_ijms, 2022, doi:10.3390/ijms24010562_

Round 1
Reviewer 1 Report
1) It is a routine paper but well written, well presented and sufficiently illustrated.
2) I believe that, at the end of the manuscript, some conclusions should be provided, explaining the contributions of this work in relation to previous knowledge.
3) Nothing else has been published in the interval 2011-2022?
4) Research Keywords do not present in a concrete way the subject covered by the work. I think that some additional keyword should be provided to avoid that the work does not fall within a field of research in pure chemistry without subsequent application.
Author Response
Thank you very much for your letter Dec. 15, 2022, concerning our manuscript titled “Title: Interfering with color response by porphyrin-related compounds in the MTT tetrazolium-based colorimetric assay”. We deeply appreciate the opportunity to submit our revised manuscript.
We carefully considered the reviewers’ comments and revised the manuscript accordingly. We also checked all references and replaced them with more relevant ones if necessary. The revised parts were highlighted in yellow in the re-submitted manuscript. We thank you for the suggestions and insights which have enriched the manuscript. The following are the revisions and responses to the comments raised by the reviewers.
Reviewer: 1
1) It is a routine paper but well written, well presented and sufficiently illustrated.
RE: We appreciate the reviewer’s complement.
2) I believe that, at the end of the manuscript, some conclusions should be provided, explaining the contributions of this work in relation to previous knowledge.
RE: According to the reviewer’s suggestion, we have added the significance and application of our study in relation to previous knowledge as below (line 253-259)
The MTT tetrazolium-based assay has been widely applied for evaluating cytotoxic effects of various agents, and many interfering factors in this assay were observed [5. 17, 18]. In the present study, we report that certain porphyrin-related compounds could induce a rapid degradation of MTT formazan under light exposure due to their photosensitizing property, resulting in significant artifacts in this assay. Accordingly, use of the MTT assay should be avoided when analyzing effects of porphyrin-related compounds and perhaps other photo-reactive compounds on cell viability. In particular, the present results should be carefully considered in the studies evaluating the cytotoxic efficacy of photodynamic agents that usually show photosensitizing property using a tetrazolium-based analysis.
3) Nothing else has been published in the interval 2011-2022?
RE: The related results have not been published since 2011. At first, we observed that the viability of cells treated with ZnPP, a heme oxygenase inhibitor, was significantly reduced based on the MTT assay without showing clear cytotoxicity. At that time, the phenomenon was thought to be due to specific action of ZnPP, which inhibits heme metabolism, and this observation did not attract our special attention for a while. Years later, we noticed that certain porphyrins including ZnPP were also strong photosensitizing compounds and suspected that these properties might be involved in decolorization of MTT formazan. In addition, publication was delayed due to the patent registration issues related to this finding. We hypothesized that MTT formazan could be used as a probe for analyzing photosensitizing activity of various compounds. After registration of the patent in 2018 regarding “FORMULATION FOR MEASURING PHOTOSENSITIZING ACTIVITY, AND KIT AND METHOD USING SAME”, we prepared to publish this manuscript and the patent-related works simultaneously.
4) Research Keywords do not present in a concrete way the subject covered by the work. I think that some additional keyword should be provided to avoid that the work does not fall within a field of research in pure chemistry without subsequent application.
RE: Thank you for the constructive suggestion. According to the reviewer’s suggestion, we have added and replaced some keywords, including porphyrin derivatives, MTT assay, interfering factors, tetrazolium, formazan decomposition, photosensitivity, photostability.
As the reviewer suggested, we have carefully reviewed our manuscript and corrected typos and other errors.

Reviewer 2 Report
The paper Interfering with color response by porphyrin-related compounds in the MTT tetrazolium-based colorimetric assay, presents an interesting study on a methodological problem about evaluation of photocytotoxicity of photosensitizers by color of tetrazolium based MTT test. However, the study was carried out with light sources such as UVC, UVA and so called fluorescence lamp in visible spectrum. These all lights are not typical for the usage with PDT and porphyrinoids compound. The results should be presented for irradiation with suitable light sources just for know how they change the photosensitizers' used and then the molecular probe for MTT assay.
Author Response
Manuscript ID: ijms-2095719
Title: Interfering with color response by porphyrin-related compounds in the MTT tetrazolium-based colorimetric assay
Thank you very much for your letter Dec. 15, 2022, concerning our manuscript titled “Title: Interfering with color response by porphyrin-related compounds in the MTT tetrazolium-based colorimetric assay”. We deeply appreciate the opportunity to submit our revised manuscript.
We carefully considered the reviewers’ comments and revised the manuscript accordingly. We also checked all references and replaced them with more relevant ones if necessary. The revised parts were highlighted in yellow in the re-submitted manuscript. We thank you for the suggestions and insights which have enriched the manuscript. The following are the revisions and responses to the comments raised by the reviewers.
Reviewer: 2
The paper Interfering with color response by porphyrin-related compounds in the MTT tetrazolium-based colorimetric assay, presents an interesting study on a methodological problem about evaluation of photocytotoxicity of photosensitizers by color of tetrazolium based MTT test. However, the study was carried out with light sources such as UVC, UVA and so called fluorescence lamp in visible spectrum. These all lights are not typical for the usage with PDT and porphyrinoids compound. The results should be presented for irradiation with suitable light sources just for know how they change the photosensitizers' used and then the molecular probe for MTT assay.
RE: We appreciate the reviewer’s valuable comments. We understand the reviewer’s concern that the evaluation of MTT formazan degradation by porphyrin derivatives was conducted under UVC, UVA and a fluorescence lamp, rather than under typical photodynamic therapy (PDT)-applicable lights e.g. LED with specific emission wavelength.
The main purpose of our study, however, was not to evaluate the PDT efficacy of porphyrin derivatives using the MTT assay, but to evaluate the interfering effects of porphyrin-related compounds with the MTT assay in general experimental conditions in regular laboratories. Since the MTT tetrazolium-based assay has been widely used for analyzing cell viability, our findings are important and meaningful in avoiding severe experimental artifacts in the relevant studies conducted in ordinary laboratories.
We also observed that only porphyrins with photosensitizing property under specific lights induced MTT formazan degradation. The reasons to use UVC, UVA and a fluorescence lamp was to prove why porphyrin-derivatives decomposed MTT formazan; this was to clarify that the results observed here were due to the photosensitivity based on the light energy absorbed by each porphyrin compound, and not the total light energy exposed. As the reviewer’s suggestion, currently we are investigating the effects of porphyrin-related compounds on the MTT formazan degradation under different sources of lights including red, blue, green, and white LED etc.
The reason and purpose of our study were described in line 69-77 of the introduction and re-emphasized in line 253-259 of the results and discussion.
Round 2
Reviewer 1 Report
I now have no objection to recommending publication of the manuscript.
Reviewer 2 Report
The authors explaination is suffiecient to know the goals of the study. This very important paper can be accepted for publication.